**1**   **Retrieval of aerosol optical depth from surface solar radiation measurements using machine**

**2**   **learning algorithms, nonlinear regression and a radiative transfer based look-up table**

**3**   **J. Huttunen[1,2], H. Kokkola[1], T. Mielonen[1], M. E. J. Mononen[3], A. Lipponen[1,2], J. Reunanen[4], A. V.**

**4**   **Lindfors[1], S. Mikkonen[2], K. E. J. Lehtinen[1,2], N. Kouremeti[5,6], A. Bais[6], H. Niska[7] and A. Arola[1]**

**5**   [1]{Finnish Meteorological Institute (FMI), Atmospheric Research Centre of Eastern Finland, Kuopio,

**6**   Finland.}

**7**   [2]{Department of Applied Physics, University of Eastern Finland, Kuopio, Finland.}

**8**   [3]{Kuopio, Finland.}

**9**   [4]{Tomaattinen Oy, Helsinki, Finland.}

**10**   [5] {Physikalisch-Meteorologisches Observatorium Davos, Dorfstrasse 33, CH-7260 Davos Dorf,

**11**   Switzerland.}

**12**   [6] {Aristotle University of Thessaloniki, Laboratory of Atmospheric Physics, Thessaloniki, 54124,

**13**   Greece.}

**14**   [7]{Department of Environmental and Biological Sciences, University of Eastern Finland, Kuopio,

**15**   Finland.}

**16**       Corresponding author: J. Huttunen, Finnish Meteorological Institute (FMI), Kuopio Unit, P.O.

**17**   Box 1627, FI-70211 Kuopio, Finland. (jani.huttunen@fmi.fi)

**18**   Key points:

**19**   -Machine learning methods can produce very good aerosol optical depth estimates from surface solar

**20**   radiation data

**21**   -These tools have the potential to be used to retrieve long aerosol optical depth time series from surface

**22**   solar radiation measurements

**Abstract**
In order to have a good estimate of the current forcing by anthropogenic aerosols knowledge on past
aerosol levels is needed. Aerosol optical depth (AOD) is a good measure for aerosol loading. However,
dedicated measurements of AOD are only available from 1990's onward. One option to lengthen the
AOD time series beyond 1990's is to retrieve AOD from surface solar radiation (SSR) measurements
done with pyranometers. In this work, we have evaluated several inversion methods designed for this
task. We compared a look-up table method based on radiative transfer modelling, a nonlinear
regression method and four machine learning methods (Gaussian Process, Neural Network, Random
Forest and Support Vector Machine) with AOD observations done with a sun photometer at an Aerosol
Robotic Network (AERONET) site in Thessaloniki, Greece. Our results show that most of the machine
learning methods produce AOD estimates comparable to the look-up table and nonlinear regression
methods. All of the applied methods produced AOD values that corresponded well to the AERONET
observations with the lowest correlation coefficient value being 0.87 for the Random Forest method.
While many of the methods tended to slightly overestimate low AODs and underestimate high AODs,
Neural network and support vector machine showed overall better correspondence for the whole AOD
range. The differences in producing both ends of the AOD range seem to be caused by differences in
the aerosol composition. High AODs were in most cases those with high water vapour content which
might affect the aerosol single scattering albedo (SSA) through uptake of water into aerosols. Our study
indicates that machine learning methods benefit from the fact that they do not constrain the aerosol
SSA in the retrieval where as the LUT method assumes a constant value for it. This would also mean
that machine learning methods could have potential in reproducing AOD from SSR even though SSA
would have changed during the observation period.

**1.Introduction**

The Fifth Assessment Report of the Intergovernmental Panel on Climate Change states that the most significant source of uncertainty in the projections of climate is related to aerosols (IPCC, 2013). One significant contribution to this uncertainty comes from the fact that without the knowledge of the aerosol burden in the past, we are not able to estimate the current forcing of anthropogenic aerosol. For example, the effect of changes in the current aerosol emissions on climate depends on the background aerosol load during the pre-industrial era (e.g. Andreae, 2008; Carslaw et al., 2013). In addition, the current estimates of past aerosol emissions are highly uncertain (Granier et al., 2011) thus increased knowledge on historical aerosol levels would increase our ability to estimate the present day aerosol radiative forcing.

One limiting factor in determining the properties of global aerosol in the past has been that observations of aerosol radiative effects have been limited to fairly recent periods. For example, the aerosol optical depth has mainly been measured using sun photometers and the most widely known ground-based network of sun photometers is Aerosol Robotic Network (AERONET; Holben et al., 1998). Although, AERONET contains globally already over 700 stations, with a fairly good spatial coverage compared to many other observation networks, it still lacks in temporal coverage, providing aerosol optical properties and AOD only since 1990s, and reaching the current status until the recent years. The earliest records of satellite-based AOD are provided by TOMS (Total Ozone Mapping Spectrometer, e.g. Torres et al., 2002) and AVHRR (Advanced Very High Resolution Radiometer, Geogdzhayev et al., 2005), from 1979 and 1983 onwards, respectively. However, neither one of these instruments were specifically designed to retrieve aerosol properties. The more recent dedicated aerosol sounders, such as ATSR (The Along Track Scanning Radiometer 2, Llewellyn-Jones and Remedios, 2012), MODIS (Moderate Resolution Imaging Spectroradiometer, Levy et al., 2010), VIISR (Visible

Infrared Imaging Radiometer Suite, Jackson et al., 2013), and MISR (Multi-angle Imaging
SpectroRadiometer, Kahn and Gaitley, 2015) offer data from 1995, 2000 and 2002 onwards,
respectively. It is therefore apparent that neither sun-photometer nor satellite records of AOD are
available for all the decades where industrialization has had a significant effect on the aerosol load.
There have been, however, recent studies where aerosol load has been indirectly retrieved from global
surface solar radiation (SSR) or separately from direct and diffuse radiation measurements, which
cover much longer time periods than sun photometer and satellite observations of AOD. Recently,
Kudo et al., 2011 and Lindfors et al., 2013 used radiation measurements done with pyranometers and
pyrheliometers to estimate AOD. Lindfors et al., 2013 demonstrated that AOD can be estimated by
using SSR and water vapor information and a look-up table (LUT) generated with a radiative transfer
code. Their method produces AOD estimates that have 2/3 of the results within ± 20 % or ± 0.05 of
collocated AERONET AODs. Because pyranometer SSR measurements have been done since 1950's
over the globe, the usage of AOD estimates based on SSR measurements would enable us to construct
AOD time series that go several decades back in time.
Since the 1990's machine learning methods have made their way to atmospheric sciences and have
been used in e.g. satellite data processing, climate modeling, and weather prediction (Hsieh, 2009).
Because of their ability to retrieve parameters from data that have strongly non-linear relationships,
they have potential of retrieving AOD from a combination of solar radiation measurements and
auxiliary data such as water vapour content (WVC) and solar zenith angle (SZA), similarly to what was
done by Lindfors et al. (2013) using a radiative transfer based approach. The aim the present work is to
investigate how well machine learning methods are able to estimate AOD from pyranometer
observations, by evaluating their performance in comparison with a radiative transfer based look-up-
table approach. We chose four different methods: Neural Network (NN, McCulloch and Pitts, 1943),
Random Forest (RF, Breiman, 2001), Gaussian Process (GP, Santner et al., 2013) and Support Vector
Machine (SVM, Smola and Schölkopf, 2004) and compared them against a look-up table and a
nonlinear regression method (NR, Bates and Watts, 1988). The performance of these methods was
evaluated with AERONET AOD observations done in Thessaloniki, Greece, after the AOD estimates
were derived with SSR observations. Nonlinear regression has been successfully used in multiple
studies within aerosol and atmospheric sciences (e.g. Huttunen et al., 2014; Ahmad et al., 2013). Out of
these machine learning methods, Neural network (NN) has been actively used in different types of
applications in atmospheric sciences. For example, it has been applied to retrieve aerosol properties
from remote sensing instruments (Olcese et al. 2015; Taylor et al., 2014; Foyo-Moreno et al, 2014).
There have been, however, recent studies where aerosol load has been indirectly retrieved from global
surface solar radiation (SSR) or separately from direct and diffuse radiation  measurements, which
cover much longer time periods than sun photometer and satellite observations of AOD. Recently,
Kudo et al., 2011 and Lindfors et al., 2013 used radiation  measurements done with pyranometers and
pyrheliometers  to estimate AOD. The study by Olcese et al. 2015 is similar to ours in the sense that
they use alternative data together with Neural Network approach in an attempt to retrieve AOD at an
AERONET site. In their study, they fill in missing AOD values (due to e.g. cloud cover) at one
AERONET station based on trajectories and AOD observed on another site. To our knowledge, the rest
of the analyzed methods have not been used to retrieve aerosol properties directly from observations.
**2. Data and Methods**
We compared the ability of several methods to estimate AOD, based on SSR and water vapor
measurements (and SZA that can be readily determined for any given time and location), against
AERONET AOD measurements at 500 nm (henceforth AOD) done at Thessalonki, Greece. This site
was chosen for this study, because it has all the necessary measurements with high quality from a 10
year time period, and because it is the same site to which Lindfors et al. (2013) applied their LUT-
approach. Furthermore, the location has varying aerosol concentrations and relatively high AOD values
throughout the year.
**2.1 Pyranometer measurements of surface solar radiation**

SSR has been measured at Thessaloniki since January 1993 with a CM21 pyranometer manufactured
by Kipp and Zonen. The instrument is located on the roof of the Physics Department at the Aristotle
University of Thessaloniki (40.63 N, 22.96 E), ca. 60 m above sea level. The data are sampled every 1–
2 s and every minute the average and standard deviation of the samples are recorded (see more details
from Lindfors et al., 2013). The calibration of the pyranometer has been confirmed to stay within the
quoted by the manufacturer accuracy (Bais et al., 2013).
**2.2 AERONET measurements**
AERONET is a network of sun and sky scanning radiometers that measure direct sun and sky radiance
at several wavelengths, typically centered at 340, 380, 440, 500, 670, 870, 940, and 1020 nm, providing
measurements of various aerosol related properties (Holben et al., 1998). From direct sun
measurements we exploited AOD and WVC data. When also sky radiance measurements are included,
more detailed aerosol properties such as single scattering albedo (SSA) and asymmetry parameter (gg)
can be retrieved (Dubovik et al., 2000). In the evaluation of the machine learning methods we used
Level 2.0 (cloud-screened and quality assured) AERONET direct sun measurements of AOD and WVC
for Thessaloniki. The Cimel sun photometer is located at the roof of the Physics Department in the
close vicinity of the pyranometer discussed above. From the inversion products, to interpret some of
our results in more detail, we used level 1.5 (cloud-screened) retrievals. However, when we selected the
data from the Level 1.5 inversion product, we applied all the other level 2.0 AERONET criteria except
for the AOD threshold. In other words, we applied otherwise the same rigorous quality control that is
required for Level 2 data, but we only relaxed the requirement for AOD at 440nm to range from 0.4 to
0.1, in order to have more reliable measurements for our data analysis.
**2.3 Cloud-screening of the pyranometer measurements and collocation with the AERONET**
**measurements**
Cloud-screening is a crucial factor in the analysis, thus only contribution of aerosols are considered, not
clouds. The SSR data was at first cloud-screened in order to ensure that only clear-sky measurements
were included in the analysis (see Lindfors et al., 2013 for more details). However, during the analysis
of the data it became evident that even after the initial cloud-screening, the SSR data still included
observations that deviated significantly from the main body of the observations. Since there is a high
probability that these outliers in the data were caused by e.g. cloud-contamination, we applied
additional screening to the data. Thus, we removed the clear outliers of possibly undetected clouds, in
our case those observations that deviated by more than $\pm20$ Wm$^{-2}$ from the exponential regression fit
(SSR = a×exp(-b×AOD)+c, where a, b and c are regression constants). This additional screening was
applied through regression of SSR against AOD for a given range of SZA (within $\pm0.5°$). It has to be
noted that this data was only a small fraction of all the data that remained after the cloud screening and
it is very unlikely that the additional cloud-screening would affect the main results and the conclusions
of our study.
The SSR values were collocated for each AOD with the $\pm1$ minutes difference, averaged and finally
normalized for the Sun-Earth distance corresponding to January 1$^{st}$. The training dataset for the
machine learning methods contained years 2009-2014 and the validation (verification) dataset years
2005-2008. These periods were selected because we wanted to verify if the methods could provide
reasonable AOD estimates for a period different than the training. The training dataset covered
approximately 2/3 and the validation dataset 1/3 of the whole data. For all the methods the input
parameters are SSR, WVC and SZA, and they produce AOD estimates (Table A1 summarises the
statistics of maximum, minimum, average, STD and median for the input and the output parameters.
Table A1 shows that e.g. AOD is larger for the validation dataset, although the maximum value is larger
for the training).

## 168    2.4 LUT and NR methods for AOD retrievals
### 170    2.4.1 Radiative transfer model based look-up table (LUT)

To retrieve AOD from SSR observations Lindfors et al., (2013) produced a LUT based on radiative
transfer simulations. They simulated SSR in different atmospheric conditions by varying AOD, WVC
and SZA systematically. They used a single aerosol model for all the simulations, and therefore called
their AOD estimate as an effective AOD, which is only a function of SSR, SZA, WVC. Other
parameters were assumed as constants e.g. Ångström Exponent of 1.1, SSA at 500 nm of 0.92 (the
SSA's spectral pattern follows the rural background aerosol model by Shettle, 1989, where SSA
changes from roughly 0.92 at 400 nm to 0.89 at 1000 nm), the asymmetry parameter was assumed
wavelength independent with a value of 0.68 while the albedo was varying with wavelength and SZA.
For a more detailed description of the LUT method see Lindfors et al., (2013).

### 180    2.4.2 Nonlinear regression method (NR)

The nonlinear regression (NR) is a multivariate analysis method which is used when the dependencies
between the study variables are not linear (Bates and Watts, 1988). NR is useful especially when there
are physical reasons for believing that the relationship between the response and the predictors follows
a particular functional form. Benefits of NR are that it needs only moderate sized sample of the studied
phenomena to give adequately precise results and as an output it gives a simple, but not predefined,
function for prediction. Additional advantage of NR against the other methods presented in this paper is
that once the parameters are estimated, they can be used in similar cases without additional training
data. In this study we assume that AOD can be estimated as a function of SSR, WVC and SZA.
Multiple different formulations for the NR function was tested and the function with the best prediction
ability found for this data is given by

$$
\begin{aligned}
\mathrm{AOD} = b_0 &+ b_1 \exp\!\left(\frac{1}{\mathrm{SZA}}\right) + b_2 \exp\!\left(\frac{1}{\mathrm{SSR}}\right) + b_3 \exp\!\left(\frac{1}{\mathrm{WVC}}\right) \\
&+ b_4 \exp\!\left(\frac{1}{\mathrm{SZA}} + \frac{1}{\mathrm{SSR}}\right) + b_5 \exp\!\left(\frac{1}{\mathrm{SZA}} + \frac{1}{\mathrm{WVC}}\right) + b_6 \exp\!\left(\frac{1}{\mathrm{SSR}} + \frac{1}{\mathrm{WVC}}\right).
\end{aligned}
\tag{1}
$$


The coefficients $b_0$-$b_6$ were determined using R-software (R Core Team, 2014) and are shown in Table
A2.
**2.5 Machine learning methods for AOD retrievals**

**2.5.1 Neural Network (NN)**
Artificial neural networks belong to the family of machine learning methods (McCulloch and Pitts,
1943). As usual in machine learning methods, the aim of an artificial NN is to generate a mathematical
model to represent the phenomenon that is examined. The mathematical model of NN structure
specifically consists of interconnected neurons with numeric weights. A typical NN model is multilayer
perceptron (MLP) (Rosenblatt, 1958), which is used in this study. A MLP network consists of several
neuron layers: an input layer, hidden layers and an output layer. The weights and other parameters of
the model are tuned or trained with a specific training data set containing input-output pairs of the
phenomenon. In this case the model inputs are SSR, WVC, SZA and the output is AOD. The training is
executed with a training algorithm and in this paper the Levenberg-Marquardt algorithm is used (Hagan
and Menhaj, 1994). A total of 20 NNs were trained in this case. The NNs differed from each other by
the number of neurons in a hidden layer. Five networks with the smallest prediction error within the
training data set were selected to the final committee of networks. The final prediction of the NN model
was computed as a median of the outputs of all networks in the committee. For more information on
NNs see, for example, Bishop, (1995).
**2.5.2 Random Forest (RF)**
Random Forest is a machine learning technique that may be used for classification and nonlinear
regression (Breiman, 2001). RF for nonlinear regression consists of an ensemble of binary regression
trees. Each of these trees is constructed using a randomized training scheme and is essentially a
piecewise constant fit to the training data set. The prediction of a RF model is obtained by averaging
the regression tree predictions over the whole model ensemble. In this study, the RF implementation
from the Scikit-Learn machine learning library (Pedregosa et al. 2011) was used. We used (SSR, WVC,
SZA, SSRxWVC, SSWxSZA, WVCxSZA) as the RF model inputs and AOD as the output. A
randomized cross-validation scheme was used to find the optimal training parameters for the RF. For
more information on RFs see, for example, Friedman et al., (2001).
**2.5.3 Support vector machine (SVM)**
Support vector machine (SVM) is a machine learning technique (Vapnik, 1995; Burges, 1998). In this
study, we use the standard SVM regression (SVR), the formulation based on the commonly used $\varepsilon$-
SVR with radial basis kernel function. For implementing the SVM the libsvm package was used
(Chang and Lin, 2011). The objective of $\varepsilon$-SVR is to find a function that has at most $\varepsilon$ deviation from
the training data set outputs. The training of an $\varepsilon$-SVR model is formulated as a quadratic (convex)
optimization problem in which the Vapnik's ε-insensitive loss function is minimized (e.g. Vapnik 1995).
The ε-SVR model has two training parameters that were used to control the training: the regularization
parameter, which controls the smoothness of the approximation function (sensitivity to noise), and the
parameter ε, which dominates the number of support vectors by governing the accuracy of the
approximation function. The determination of SVM control parameters was solved by the means of a
grid search. For a more detailed description of the method, the reader is referred, for example, to Smola
and Schölkopf (2004).
**2.5.4 Gaussian process (GP)**
Gaussian process (GP) for machine learning is a generic supervised learning method that may be used,
for example, for nonlinear regression. In GP learning, the function inputs and outputs are treated as
Gaussian random variables and the correlations between these variables are modelled. The predictions
given by a GP model are computed as conditional probability distributions given the training data and
function inputs. As the prediction given by a GP model is a probability distribution, the error estimates
for the predicted point estimates are obtained automatically. In this study, the GP implementation from
the Scikit-Learn machine learning library was used. The same inputs and output variables as with the
RF models were used in the GP training. The best performing correlation function training parameters
were sought for using maximum likelihood estimation. A total of 25 GP models were trained. The
training of each model was carried out using 2500 training data samples that were randomly sampled
from the full training data set. The five best performing GP models were selected into the final GP
model committee. The final prediction was computed as the median of the predictions given by the GP
models in the committee. For more information on GPs for machine learning see, for example, Welch
et al., (1992), Rasmussen and Williams (2006), and Santner et al., (2013).

## 3. Results

### 3.1 Comparison of the methods

Table 1 shows the statistics of the AOD observed by AERONET together with the statistical characteristics of the predicted AOD for the years 2005-2008. From the table, we can see that predicted values show good correlation against the observations for all the methods. Predictions by RF had the lowest correlation coefficient with a value of 0.87 while the correlation coefficient for NR was only slightly larger, 0.88. For the best performing methods, LUT, GP, NN, and SVM, the correlation coefficients were approximately 0.92. Their predicted AODs in comparison to AERONET AOD are shown in Figure 1. To visualize the distribution of the data, the colorbar in Figure 1 represents the number of observations for each AOD interval of 0.005. Based on the different statistics in Table 1, machine learning methods (NN, SVM, GP) produce a good match with AERONET data and they perform equally good or better than the LUT method according to all the metrics. Due to the fact that RF and NR are not able to produce as good estimates as the LUT method, they were left out from the more detailed analysis.

Although these methods are able to predict the average AOD with a good accuracy, they differ when we compare their ability to predict different AOD levels. In Figure 1, the color scales indicate the absolute amount of results in the areas with the interval of 0.01x0.01 (vertically and horizontally) for AOD, in addition 1:1-lines and linear fits are included. Based on the linear fits, NN appears to have the best agreement with AERONET data for the whole AOD range. As the average and median values of AERONET AOD are 0.240 and 0.207 respectively (Table 1), the main population of the measurements is in the range of moderate AODs. The machine learning methods are obviously weighted to perform best in this range of AODs. However, from Figure 2, which shows the absolute difference between

AERONET and predicted AOD, we can see that LUT and GP tend to significantly underestimate AOD
for AODs larger than 0.5, while NN and SVM are able to reach smaller differences with AERONET on
average, although with larger overall variabilities than LUT and GP. Although NN and SVM also start
to deviate from the observations at higher AODs, these deviations are more modest in relative sense as
can be seen from Figure 3 which shows the relative difference between the observations and
predictions. All the methods overestimate AOD in relative terms, when AOD approaches zero (Figure
3). However, as Figure 2 demonstrates, the absolute error is systematically very low in the small AOD
region (AOD < 0.2). NN and SVM are generalized better for large AODs than the other methods,
where the amount of data are small.
As an additional test, we tested combinating differenthe colorbar in Figure 1 represents the number of
observations for each AOD interval of 0.005. methods. In Table 1, the four last rows represent the
values for cases where the results of machine learning methods are combined by averaging them. As
can be seen from the table, these combinations do not improve the estimates compared to the statistical
values of individual methods.
**3.2 The effect of water vapour on AOD predictions**
Huttunen et al. (2014) showed that WVC and AOD have typically a positive correlation. Therefore, we
investigated how the AOD estimates from different methods are affected by WVC. Figure 4 shows the
relative difference between the predictions and measured AOD with respect to WVC. From this figure,
we can see that the LUT-based AODs are overestimated at the smallest and underestimated at the
largest WVC contents. The reason for this behaviour is that the LUT method has been set to assume
prescribed and constant properties for many relevant parameters that affect SSR (other than AOD and
WVC); e.g. aerosol single scattering albedo, asymmetry parameter and surface albedo (Lindfors et al.,
2013). Consequently, the assumption of constant SSA in particular leads to WVC-dependent systematic
bias of the LUT-based AOD, as we will show next. The other methods are closer to the ratio of 1
without such a systematic bias, excluding the SVM's underestimation for the smallest WVC.
Figure 5 shows measured SSR and LUT-based SSR for a narrow set of SZAs (48.50°-51.50°). AOD is
on the horizontal axis, SSR on the vertical axis and WVC is shown with the colorbar. From Figure 5a it
is evident that LUT incorporates a strong WVC-dependent structure: for a given SSR level, AOD
decreases with increasing water vapor content. This pattern follows from the assumption that the
aerosol composition remains the same, i.e. it has a fixed SSA value. Thus in the LUT method, increases
in SSR absorption by water vapour are compensated by decreases in aerosol extinction. In the real
atmosphere, water vapour content has also implications on aerosol composition and size. If all
conditions apart from water vapour remained constant, increase of water vapour would also increase
the uptake of water into aerosol particles thus affecting the aerosol SSA. The effect of fixed SSA is also
visible in the way the LUT-based AOD estimates are distributed (Figure 5a). In Figure 5c we can see
that for a given AOD in the LUT, the highest WVC values always correspond to the lowest SSR values.
However, the same pattern is not clearly visible either in the plot with the measured values (Figure 5b)
or in the plot with AOD from NN (Figure 5d). This indicates that although the machine learning
methods do not get explicitly any information about the possible systematic co-variability of WVC and
SSA, they seem to be able to detect it indirectly, at least to some extent.
To further illustrate this, Figure 6a shows the AERONET measurements of AOD and single scattering
co-albedo, 1-SSA at 500 nm as a function of WVC. Here, together with the absorption strength by the
water vapour, we considered more illustrative to show the single scattering co-albedo rather than SSA.
In this plot, SZA, SSR and season were limited respectively to: $58° < SZA < 62°$, $420$ Wm$^{-2}$ < SSR <
$460$ Wm$^{-2}$, June-August, allowing enough data with the limited parameters. Thus, the plot illustrates the
co-variability of WVC and SSA for a limited range of surface solar radiation and SZA, for conditions
when the LUT method produces lower AOD values for higher WVC (Figure 5a). However, Figure 6a
clearly shows that an opposite relationship between AOD and WVC is obtained by the measurements.
Moreover, this pattern is compensated by aerosol absorption (remember that in this sub-set we
constrained SSR), which decreases with increasing WVC; this is likely related to the aerosol swelling
by hygroscopic growth that increases the scattering of the aerosol. Therefore, we can conclude from the
measurements that because of the co-variability of WVC and SSA in Thessaloniki, the assumption of a
fixed SSA in the LUT causes limitations for predicting AOD, while the machine learning methods can
take into account, at least to some extent, this relationship indirectly. Using radiative transfer modeling
we demonstrated that the magnitude of these changes in water vapor and aerosol absorption, as
indicated in Figure 6. Indeed, they induced opposite effects of similar magnitude in surface solar
irradiance. For the base case, we simulated SSR with WVC of 2.8 cm and 1-SSA of 0.06 (with SZA of
60° and AOD of 0.3) as inputs, resulting in 439.9 $Wm^{-2}$. When we increased the water vapour column
to 3.6 cm, the corresponding decrease in SSR was about 6.8 $Wm^{-2}$. However, when we additionally
decreased the aerosol absorption (1-SSA) to 0.04, the difference to the base case shrank to 1.8 $Wm^{-2}$
and this remaining amount can mostly be explained by the asymmetry parameter, which also exhibits a
systematic dependence with WVC (stronger forward scattering by particles grown in humid
conditions).
The lower panel of Figure 6 further illustrates the role of fixed SSA in the observed WVC-dependent
bias in the LUT results, which can be avoided with the machine learning methods. It shows the mean
ratio of LUT-estimated and AERONET-measured AOD on the right-hand side y-axis as a function of
water vapour content (so essentially the same results shown by a box-plot in Figure 4). Additionally, on
the left-hand side y-axis, the single scattering albedo (estimated for 500 nm) from AERONET
measurements is shown as a function of water vapour amount as well. This also demonstrates that the
over- and underestimations of the LUT method coincide with SSA range that is under and over the
assumed fixed value of 0.92 (shown with red dashed line), respectively. Visibly, the ratio in the right-
hand axis of Fig. 6b, reaches one not until SSA is roughly 0.93 instead of 0.92. Presumably, SSA has
actually a different wavelength pattern than the one assumed in LUT.
**4. Conclusions**
We have used several inverse methods to retrieve aerosol optical depth (AOD) from surface solar
radiation (SSR) and water vapour content (WVC) measurements (with corresponding solar zenith angle
data) done in Thessaloniki, Greece. Two traditional (look-up table (LUT) and nonlinear regression
(NR)) and four machine learning methods (Gaussian Process (GP), Neural Network (NN), Random
Forest (RF) and Support Vector Machine (SVM)) were used to retrieve AOD estimates for the years
2005-2008. Then we compared the AOD estimates with collocated AOD measurements done by
Aerosol Robotic Network (AERONET). Our comparisons showed that:
-AOD estimates based on the LUT method agreed better with AERONET than the NR estimates but
apart from RF, the machine learning methods produced AOD estimates that were comparable or better
than LUT.
-NN and SVM methods reproduced good correspondence to AERONET observations for both low and
high AODs while rest of the methods tended to overestimate low AODs and underestimate high AODs.
The main reason for the better performance of these machine learning methods was that there were no
constrains of the aerosol single scattering albedo (SSA) in the retrieval. In other words, the methods do
not need to explicitly make assumptions on the optical aerosol properties of the atmosphere and
because seem to be able to indirectly account for the covariation of WVC and SSA.
-When compared with AERONET measurements, the best AOD estimates were retrieved with the
machine learning algorithms, but only NN and SVM were able to generalize accurate estimates also for
large AODs.
-The machine learning methods are sensitive to the selection of the training data set and other
constraints, and are generally valid only for the range of the variables used for their training; thus care
needs to be taken when these methods are employed.
-These tools have the potential to be used in the retrieval of AOD from SSR measurements to lengthen
the time series of AOD. Historical AOD is essential in the estimation of anthropogenic aerosol effects
and in the evaluation of AOD retrievals from space borne instruments before the 1990s.
-The intention of comparing different methods was to test their ability in an "out-of-the-box"
configuration. With this in mind, methods were not particularly tuned to reach the best possible results.
It is very likely that e.g. optimizing the free parameters used in each of the nonlinear modeling
approaches, their ability to reproduce observed AOD could be further improved.
**Acknowledgements.**
We thank the AERONET team, principal investigators and other participants for their effort in
establishing and maintaining the network. This study is supported by Graduate school in Physics,
Chemistry, Biology and Meteorology of Atmospheric Composition and Climate Change: From
Molecular Processes to Global Observations and Models. The Academy of Finland Center of
Excellence program (project number 272041) is also acknowledged. The financial support by the
strategic funding of the University of Eastern Finland is gratefully acknowledged. The author thank
Juha Tonttila and Mikko Pitkänen from Finnish Meteorological Institute, Kuopio, for their help with
the python (python.org) and in the production of the MatLab (mathworks.com) boxplot-figures. Also
JH thank the Finnish Cultural Foundation, North Savo Regional fund.

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

Table 1: Statistical characteristics of observed (AERONET) and predicted AOD by the methods of NR
(nonlinear regression), LUT (look-up table), NN (neural network), RF (random forest), GP (gaussian
process), SVM (support vector machine) and some of their combinations (averages without weights,
e.g. NN, SVM combination is their average result). Correlation coefficient ($R^2$), mean absolute
deviation (MAD), median and their ±20% percentiles between the observed and predicted. Also time
consumptions with a recent average computer power of the methods for training / estimation in the
magnitude of seconds, minutes and hours. The number of observations is 10684.

| Method | Average(std) | $R^2$ | MAD | Median | Fraction in +/-20% | Time consumption |
|---|---|---|---|---|---|---|
| AERONET | **0.240(0.147)** | | | **0.207** | | |
| NR | 0.228(0.123) | 0.880 | 0.053 | 0.210 | 48.4 % | seconds / < second |

| | | | | | | |
|---|---|---|---|---|---|---|
| LUT | 0.254(0.136) | 0.920 | 0.046 | 0.236 | 52.6 % | hours / minutes |
| NN | 0.251(0.156) | 0.920 | 0.044 | 0.212 | 59.1 % | hours / < second |
| RF | 0.225(0.116) | 0.870 | 0.052 | 0.204 | 52.9 % | tens of seconds / < second |
| GP | 0.240(0.130) | 0.927 | 0.041 | 0.213 | 60.8 % | minutes / tens of seconds |
| SVM | 0.242(0.150) | 0.918 | 0.044 | 0.201 | 58.4 % | tens of seconds / < second |
| NN, SVM | 0.247(0.152) | 0.924 | 0.043 | 0.207 | 59.7 % | |
| NN, SVM, RF | 0.240(0.138) | 0.922 | 0.042 | 0.205 | 59.9 % | |
| SVM, RF | 0.234(0.131) | 0.913 | 0.044 | 0.202 | 58.0 % | |
| NN, RF | 0.238(0.134) | 0.916 | 0.043 | 0.207 | 59.0 % | |













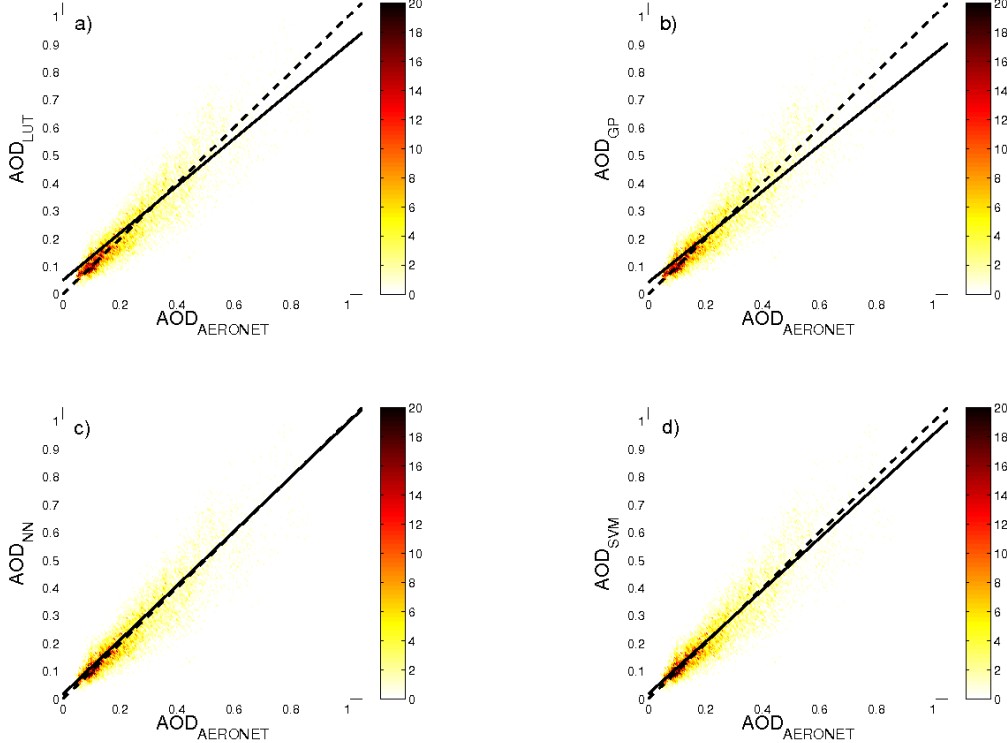

Figure 1: Observed (AERONET) and predicted AOD by the methods of a) LUT (look-up table), b) GP
(Gaussian Process), c) NN (Neural Network) and d) SVM (Support Vector Machine). The colorbar
indicates the absolute amount of results in the areas with the interval of 0.01x0.01. The 1:1-lines and
linear fits included. The number of observations is 10684. The relation for the linear fits is, estimated
AOD = a1+a2×AERONET AOD, and the coefficients of the fits are (a1, a2): 0.0503, 0.8492; 0.0429,
0.8204; 0.0164, 0.9791 and 0.0178, 0.9355, for LUT; GP; NN and SVM, respectively.


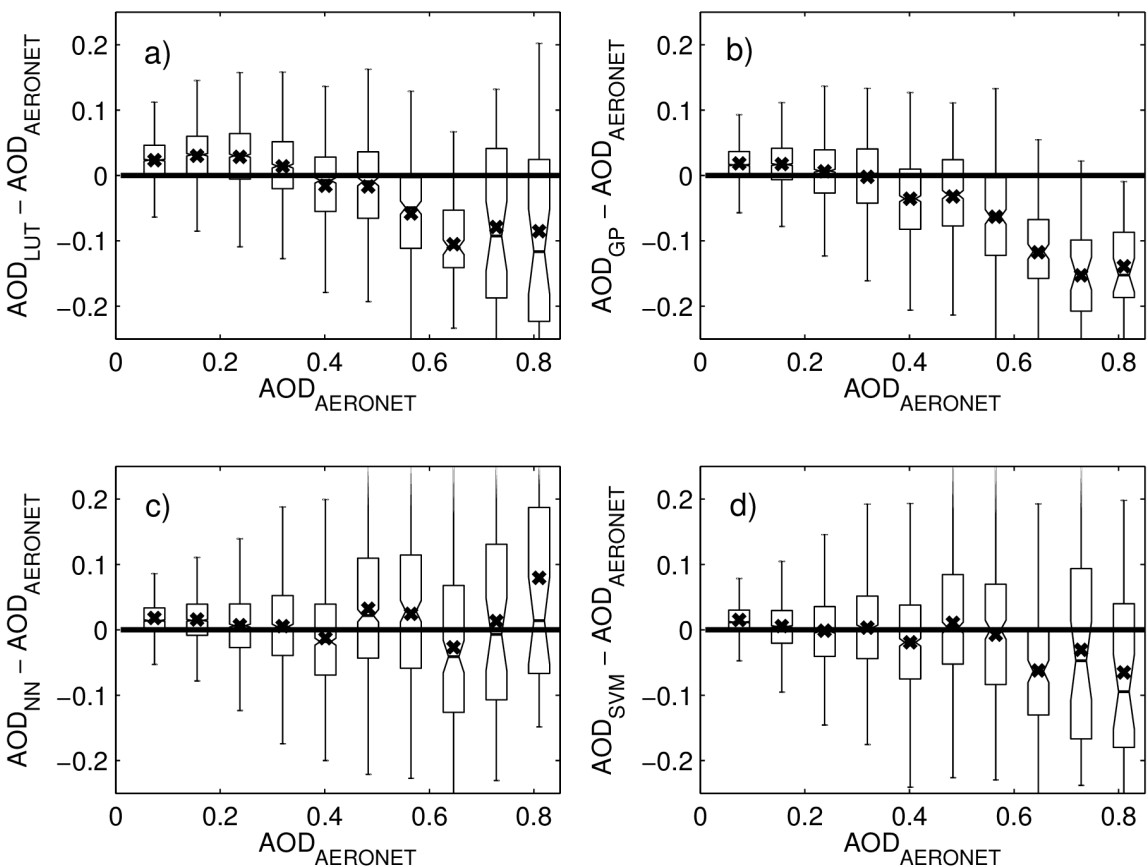

Figure 2: Differences between predicted and observed (AERONET) AOD for the methods: a) LUT
(look-up table), b) GP (Gaussian Process), c) NN (Neural Network) and d) SVM (Support Vector
Machine) with respect of the observed AOD. The crosses indicate the means of each sub-group, the
limits of the boxes are 25 %, 50 % and 75 % of the data, and the lines are plotted with 1.5 times the
inter-quartile ranges.




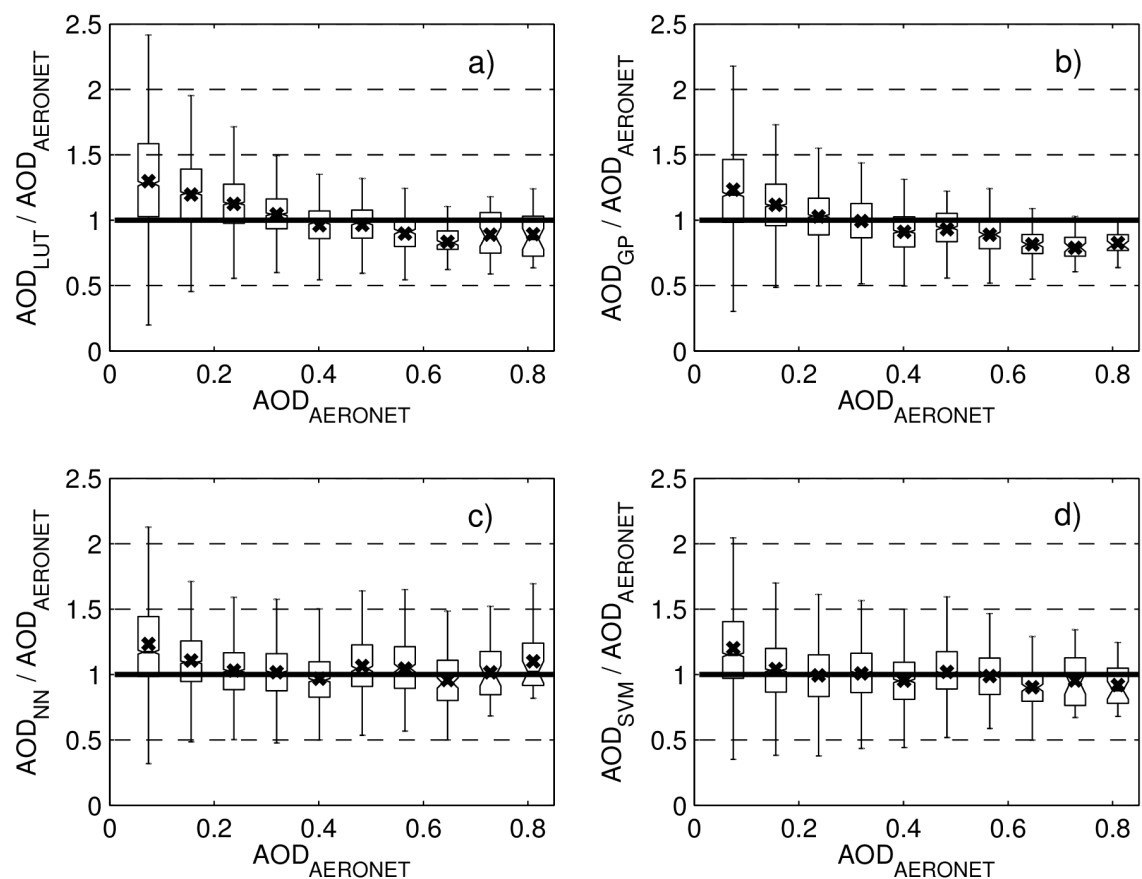


Figure 3: The same as Fig. 2, but in the vertical axis is the ratio of the predicted to the observed
(AERONET) AOD.






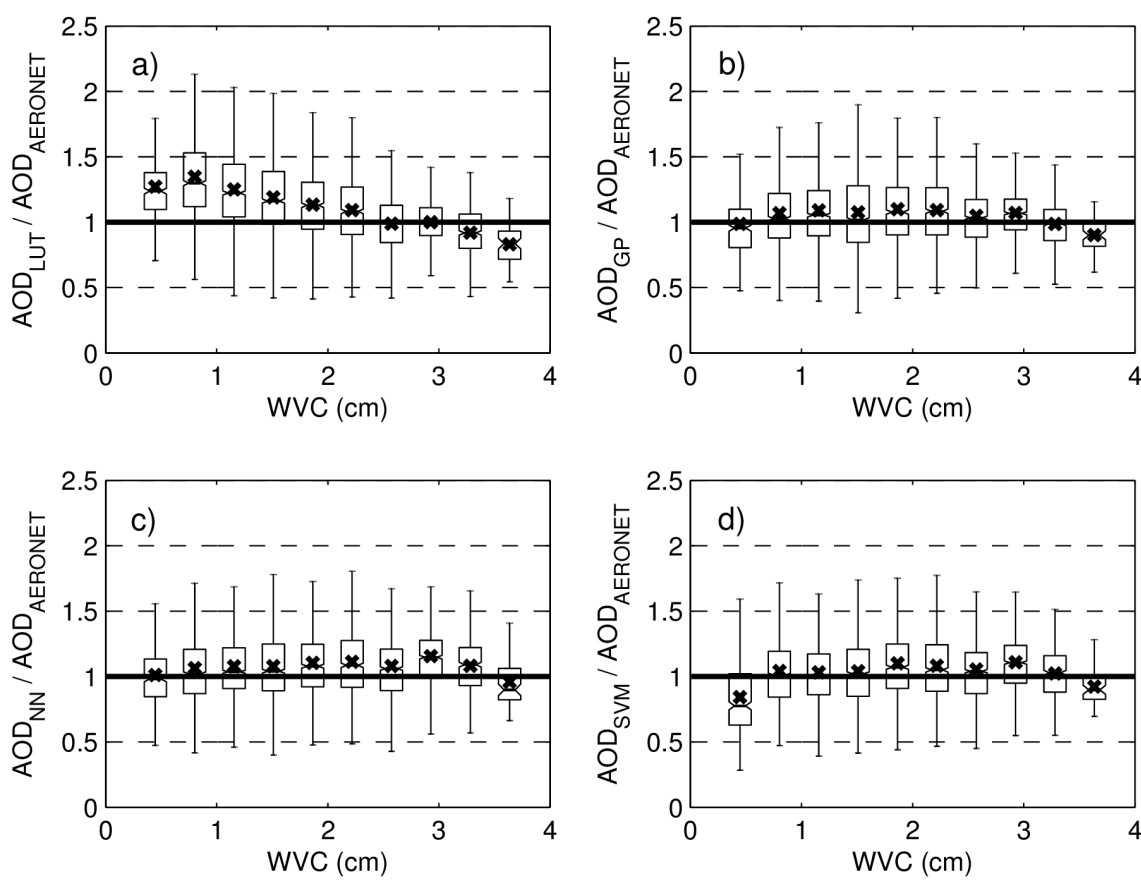


Figure 4: The same as Fig. 3, but the ratio of predicted to measured AOD is given as a function of the
water vapor column (WVC).








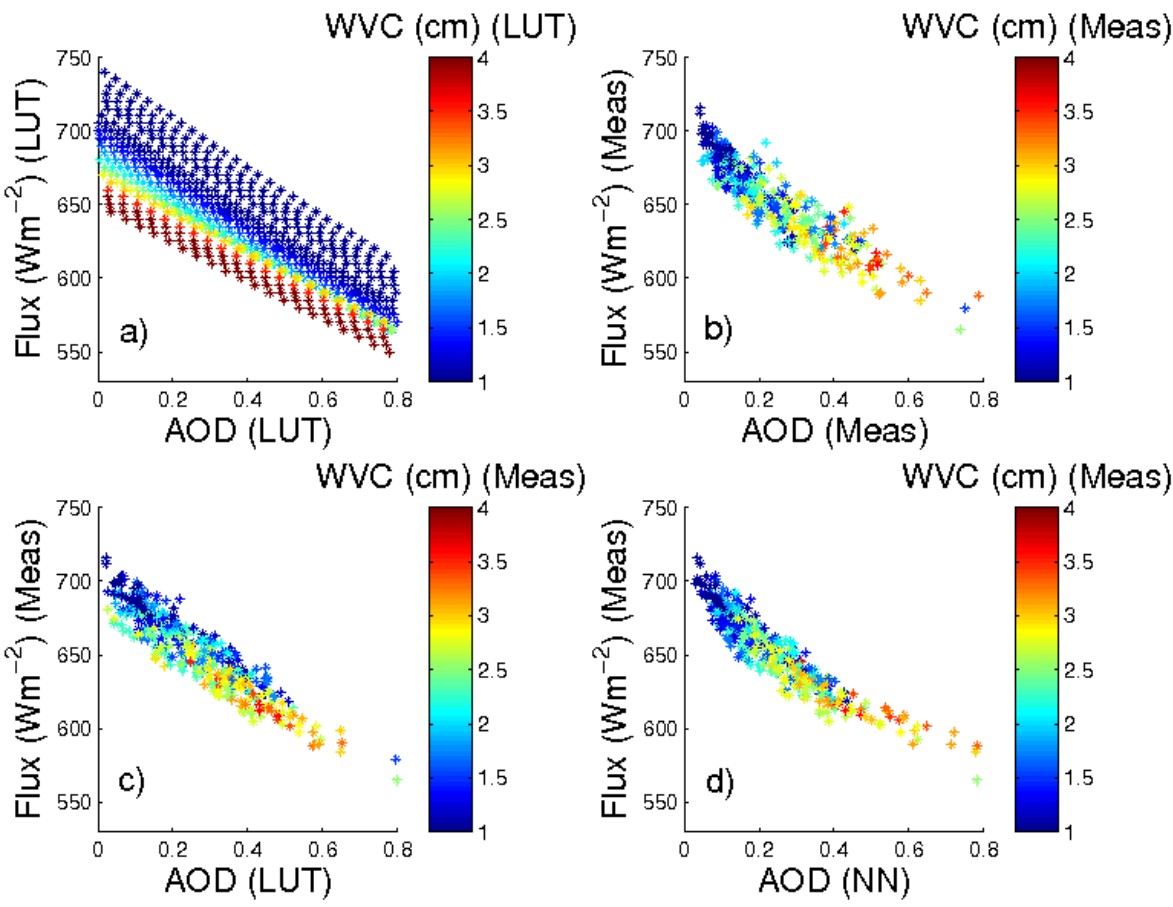


Figure 5: Solar surface radiation (SSR), aerosol optical depth (AOD) and water vapor column (WVC)

for a fixed solar zenith angle (48.50°-51.50°) for a) look-up table (LUT) and b) measurements (Meas).

The predicted AODs for c) LUT and d) neural network (NN) corresponding the same SSR, WVC and

SZA.

555

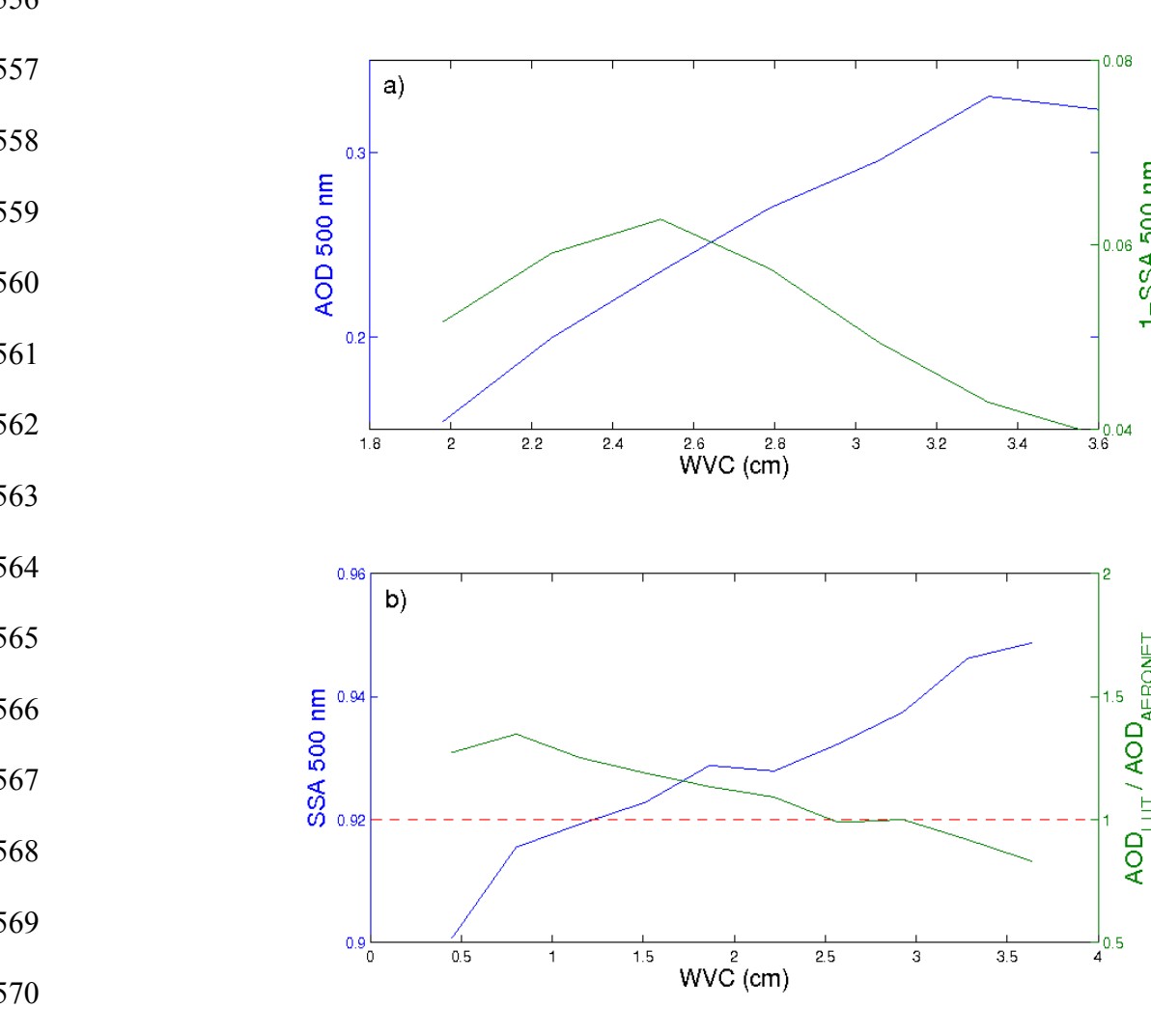

Figure 6: a) Aerosol optical depth (AOD), water vapor column (WVC) and 1-SSA at 500 nm from the

AERONET inversion sky data. b) SSA at 500 nm, WVC and the LUT's predicted AOD divided with

the observational AOD (AERONET), with the red line fixed to SSA (500 nm) = 0.92 (as in LUT).

**Appendix A**

Table A1. The statistics between the training and the validation data for the input and the output
parameters. The units for SZA, SSR and WVC are degrees, Wm$^{-2}$ and cm, respectively.

| Training: | | | | | |
|---|---|---|---|---|---|
| Parameter | Max | Min | Average | STD | Median |
| SZA | 78.6 | 17.5 | 56.2 | 15.7 | 60.0 |
| SSR | 1071.9 | 120.5 | 522.7 | 247.1 | 479.6 |
| WVC | 4.12 | 0.23 | 2.23 | 0.73 | 2.29 |
| AOD | 1.06 | 0.01 | 0.22 | 0.12 | 0.20 |
| Validation: | | | | | |
| Parameter | Max | Min | Average | STD | Median |
| SZA | 78.7 | 17.5 | 60.6 | 14.7 | 65.3 |
| SSR | 1060.0 | 113.2 | 450.2 | 235.9 | 384.5 |
| WVC | 3.81 | 0.27 | 1.87 | 0.82 | 1.79 |
| AOD | 0.85 | 0.03 | 0.24 | 0.15 | 0.21 |


Table A2.  The coefficient values of eq.(1) and errors (STD) for the NR method.

| Coefficients | Estimate | STD error | |
|---|---|---|---|
| $b_0$ | $1.716 \times 10^5$ | $8.372 \times 10^2$ | |
| $b_1$ | $-1.696 \times 10^5$ | $8.272 \times 10^2$ | |
| $b_2$ | $-1.715 \times 10^5$ | $8.363 \times 10^2$ | |
| $b_3$ | $-1.206 \times 10^1$ | $5.727 \times 10^{-1}$ | |
| $b_4$ | $1.694 \times 10^5$ | $8.264 \times 10^2$ | |
| $b_5$ | $5.145 \times 10^0$ | $2.465 \times 10^{-1}$ | |
| $b_6$ | $6.819 \times 10^0$ | $3.728 \times 10^{-1}$ | |
