# Peer review of "Retrieval of aerosol optical depth from surface solar radiation measurements using machine 2 learning algorithms, nonlinear regression and a radiative transfer based look-up table"

_Atmospheric Chemistry and Physics, 2016_

## Referee Comment (RC1) · Anonymous Referee #1 · 22 Feb 2016

General comments: The paper has interest considering the relevance to obtain aerosol optical depth (AOD) from available measurements such as solar radiation measurements. In this sense, the authors have compared several methods to estimate AOD from solar radiation measurements considering additional variables (solar zenith angle and water vapor content).

Particular comments:

1) In the paper the term "surface solar radiation" is mentioned the first time using the acronyms SSR. In order to avoid confusion is necessary to specify that it refers to

global irradiance (not direct or diffuse solar irradiance).

2) In section 1 (Introduction), the method of Foyo-Moreno et al. (2014) is mentioned along with the machine learning methods, but this method estimates AOD from solar radiation measurements using a linear relationship between AOD and a ratio. The neural network has been used to confirm the most adequate variables to take into account in the model. This should be clarified.

3) I consider that the criterion used by the authors to eliminate clouds is arbitrary or subjective in nature. Additionally, the criterion uses a function of SSR with AOD for a given solar zenith angle. What solar zenith angle? Is there then a different relationship for every solar zenith angle? The authors should use other methods, considering that there are several standard methods such as that of Long and Ackerman (2000), an automated method to identify periods of clear skies using solar radiation measurements. On the other hand, the authors assume a priori a dependence between SSR and AOD and this the task of the paper: evaluating and comparing various methods with an additional variable (water vapour content-WVC-)

4) On page 7 where the nonlinear regression method (NR) is described there is an equation with different variables, and one of them is 'flux'. Variables should be mentioned consistently; I suppose that this is Global Irradiance (SSR). On the other hand, in a paper the equation should be numbered. Also the coefficients should be specified together with their errors.

5) I don't understand paragraph 10 on page 9, with the terms used theta=, theta1L, thetaU, nugget. The same comment can be made regarding the explanation of the Random Forest method (min_samples_split, etc). In short, the machine learning methods are not clearly explained.

6) In section 3.1, in Table 1, what are the four last rows?

7) In Figure 1 the fitting equation should be included.

8) In Figure 1 I don't understand the mean of the colorbar because I think the colors should not be superimposed. The authors should clarify this.

9) In order to study the effect of water vapour content on AOS predictions, Figure 5 shows measurements of SSR versus AOD considering different values of WVC, but for a limited range of solar zenith angles (40.75o-50.25o). Why precisely this selection and not another? And how it may affect the results for other angles?

10) The pattern followed by WVC and AOD (Figure 5.a) is different from the positive correlation found by Huttuen et al. (2014).

11) Figures 5 b and 5c show no clear differences between them.

12) In their analysis, the authors have used the single scattering albedo at 550 nm, but in Figure 6 a they use the albedo for another wavelength, why?

13) Figures 6a and 6b should use the same scale for the same variable (water vapor column) in order to enable comparison. On the other hand, in Figure 6a the pattern shown for the albedo with WVC is different depending on the interval considered for the WVC (slopes with contrary signs), thus there is no consistency between Figures 6a and 6b because the pattern followed by WVC in Figure 6b is independent of the range considered at WVC. It More discussion is necessary about the effect of water vapour, considering other solar zenith angles for example.

Concluding remarks: the paper can be accepted for publication after these comments are taken into consideration and addressed.

---

## Referee Comment (RC2) · M. Taylor (Referee) · 1 Mar 2016

GENERAL COMMENTS

I read the manuscript with interest, especially considering that it performs a comparison of several multivariate techniques for modeling/estimating aerosol optical depth (AOD) using surface solar radiation (SSR) measurements. As the authors point out, long time series of such measurements are available and this can be exploited to reconstruct a coincident record also of AOD. Extrapolation of AOD back in time is something that will be very useful in studies of radiative forcing but also climate change trends. The

availability of long time series of AOD estimates will also help enrich models of other atmospheric variables that would benefit from inclusion of this important parameter. The study of AOD in the context of SSR is a very active field (a CrossRef metadata search with +"aerosol optical depth" +"solar radiation" with the "journal article" flag on returns a large number of 953,336 results), and it is good to see a study that is targeted at AOD retrieval in particular. The authors idea of comparing machine learning models is timely, well grounded and relevant to the scope of the journal of Atmospheric Chemisty and Phyics (ACP). Several of the authors were instrumental in a recent ACP paper to derive effective AOD from pyranometer measurements of SSR, by comparing the capabilities of several modern approaches, the submitted manuscript builds on this work and provides a useful feasibility study for the ballpark accuracy of AOD retrievals from irradiances using advanced models.

Methodological issues:

1) On Page 4, lines 7-9, the authors describe how they have chosen to compare neural network (NN), random forest (RF), Gaussian Process (GP) and Support Vector Machine (SVM) models of the AOD against look-up table (LUT) and nonlinear regression models. Comparative studies of this type are becoming more popular in the literature, but it should be born in mind that results are sensitive to model specification and, in particular, the number of free parameters (e.g. Ljung, 1998). For example, in the context of NN architectures alone, these include the number of neurons in hidden layers, the number of such layers, training:validation data partition sizes, neuron activation functions used). It is also rather challenging to find optimal values for model parameters. For example, Meyer et al (2003) compared a SVM alone against 16 classification methods and 9 regression methods in R. The same could be said for all of the methods adopted in the submitted manuscript. With this in mind it would be good if the authors could either:

a) increase the depth of the study by performing a thorough sensitivity analysis on the free parameters used in each of the nonlinear modeling approaches (NN, RF, GP, SVM,

and NR) to help constrain the optimal values and number of free parameters needed to achieve different model performance, or

b) emphasize more how the study performs a feasibility type of analysis of the specific nonlinear models adopted for producing AOD retrievals of certain quality.

2) On Page 6, lines 8-11, the authors describe how the training dataset for the machine learning methods contained years 2009-2014 and the validation (verification) dataset contained the previous years 2005-2008. I would like to see the authors describe why this partition was chosen (over others) as well as a short presentation of the basic exploratory statistics of these datasets: i.e. the means and standard deviations and min-max values of the model input and output parameters. This will help the authors to make stronger claims about the generality of the models selected.

SPECIFIC COMMENTS

I would say that the level of technical English in the submitted manuscript is reasonably good, as is the level of scientific description. A couple of minor points:

3) On Page 3, lines 6-7, I disagree that AERONET has rather good spatial coverage. Even on a global grid of 1 degree resolution (180 x 360 pixels), the occupancy of global pixels, is extremely low dispite there being of the order of 10ˆ3 sites.

4) On Page 3, line 15, I would say that the (satellite and AERONET AOD) records extend a between 1 and 2 decades into the past. On the daily timescale, this could be arguably be considered to be a fairly long time-series record.

5) I would make the font size bigger in Figure 1 and Figure 6.

6) In Figure 5, colour is associated with WVC and the title would be better placed vertically on the colour bars as "WVC [cm] (LUT)" and "WVC [cm] (meas.)" or something along these lines.

CONCLUDING REMARKS

Given the importance of accurate AOD estimation and the potenial for increasing the capacity for monitoring long-term changes in climate forcing where AOD is a key parameter, the submitted manuscript is a useful addition to the literature and would benefit I hope from these minor revisions.

REFERENCES

Ljung, L., 1998. System identification (pp. 163-173). Birkhäuser Boston.

Meyer, D., Leisch, F. and Hornik, K., 2003. The support vector machine under test. Neurocomputing, 55(1), pp.169-186.
* * *

---

## Author Comment (AC1) · 2 May 2016

General comments: The paper has interest considering the*

*relevance to obtain aerosol optical depth (AOD) from available measurements such as solar radiation*

*measurements. In this sense, the authors have compared several methods to estimate AOD*

*from solar radiation measurements considering additional variables (solar zenith angle and water*

*vapor content).    Particular comments:*

*1) In the paper the term "surface solar radiation" is mentioned the first time using the*

*acronyms SSR. In order to avoid confusion is necessary to specify that it refers to global irradiance*

*(not direct or diffuse solar irradiance).*

• We have modified this part as follows: "There have been, however, recent studies where aerosol load has been indirectly retrieved from global surface solar radiation (SSR) or separately from direct and diffuse radiation  measurements, which would cover much longer time periods than sun photometer and satellite observations of AOD. Recently, Kudo et al., 2011 and Lindfors et al., 2013 used radiation  measurements done with pyranometers and pyrheliometers to estimate

AOD."

*2) In section 1 (Introduction), the method of Foyo-Moreno et al. (2014) is mentioned*

*along with the machine learning methods, but this method estimates AOD from solar*

*radiation measurements using a linear relationship between AOD and a ratio. The*

*neural network has been used to confirm the most adequate variables to take into*

*account in the model. This should be clarified.*

- We changed the reference to Olcese et al., 2015, "A method to estimate missing AERONET AOD values based on artificial neural networks*"*, which is a better example of a study where Neural Networks are used for retrieving AOD. In their study, they fill in missing AOD values (due to e.g. cloud cover) at one AERONET station based on trajectories and AOD observed on another site.

*3) I consider that the criterion used by the authors to eliminate clouds is arbitrary or*

*subjective in nature. Additionally, the criterion uses a function of SSR with AOD for a*

*given solar zenith angle. What solar zenith angle? Is there then a different relation-*

*ship for every solar zenith angle? The authors should use other methods, considering*

*that there are several standard methods such as that of Long and Ackerman (2000),*

*an automated method to identify periods of clear skies using solar radiation measure-*

*ments. On the other hand, the authors assume a priori a dependence between SSR*

*and AOD and this the task of the paper: evaluating and comparing various methods*

*with an additional variable (water vapour content-WVC-)*

- The initial cloud screening was done using a similarly sophisticated method (Lindfors et al.,

2013) as that of Long and Ackermann (2000). However, after the initial screening, the remaining data still included clear outliers which were suspected to be cloud contaminated. As a

"safety precaution", we further screened the data to exclude these outliers. It has to be noted that the excluded data was only a small fraction of all the data that remained after the cloud screening and it is very unlikely that the additional cloud-screening would affect the main results and the conclusions of the study. Therefore, we feel that an alternative cloud-screening method would not change our main results and conclusions. This is clarified in the revised manuscript.

*4) On page 7 where the nonlinear regression method (NR) is described there is an*

*equation with different variables, and one of them is 'flux'. Variables should be men-*

*tioned consistently; I suppose that this is Global Irradiance (SSR). On the other hand,*

*in a paper the equation should be numbered. Also the coefficients should be specified*

*together with their errors.*

• We have rewritten the equation in the revised manuscript as follows:

$$AOD = b_0 + b_1 \exp\left(\frac{1}{SZA}\right) + b_2 \exp\left(\frac{1}{SSR}\right) + b_3 \exp\left(\frac{1}{WVC}\right)$$
$$+ b_4 \exp\left(\frac{1}{SZA} + \frac{1}{SSR}\right) + b_5 \exp\left(\frac{1}{SZA} + \frac{1}{WVC}\right) + b_6 \exp\left(\frac{1}{SSR} + \frac{1}{WVC}\right).$$

• In addition, we have included the coefficient values and errors. See Table A2 in the revised manuscript.

*5) I don't understand paragraph 10 on page 9, with the terms used theta=, theta1L,*

*thetaU, nugget. The same comment can be made regarding the explanation of the*

*Random Forest method (min_samples_split, etc). In short, the machine learning meth-*

*ods are not clearly explained.*

● The descriptions of the methods are now updated in the section "**2.5 Machine learning**

**methods for AOD retrievals"**. The machine learning model descriptions were homogenized. A

sentence on how the selection of the training parameters was carried out was added for each of the models.

*6) In section 3.1, in Table 1, what are the four last rows?*

● In Table 1, the four last rows represent the values for cases where the results of machine learning methods are combined by averaging them. This is now clarified in the manuscript.

*7) In Figure 1 the fitting equation should be included.*

● The fitting equation is now presented in the figure caption.

*8) In Figure 1 I don't understand the mean of the colorbar because I think the colors*

*should not be superimposed. The authors should clarify this.*

● The colorbar represents the number of observations for each AOD interval of 0.005. The reason why the colorbar was included is that it helps the reader visualize the distribution of AOD

values. We have clarified this in the revised manuscript.

*9) In order to study the effect of water vapour content on AOS predictions, Figure 5*

*shows measurements of SSR versus AOD considering different values of WVC, but*

*for a limited range of solar zenith angles (40.75o-50.25o). Why precisely this selection*

*and not another? And how it may affect the results for other angles?*

• Here, we selected the SZA range so that we get enough data for the analysis on the other hand keeping the range as narrow as possible. The purpose here was to see how LUT handles the

AOD estimation especially with respect of AOD and WVC compared with the measurements and machine learning methods. The effect is difficult to see, that is why, we updated the figure in the revised manuscript with a larger SZA range (48.50-51.50 degrees, instead of 49.75-50.25

degrees). Now, the figure is clearer and evidently LUT is in problems whereas NN handles the observed pattern better. The essential result holds also for other SZAs.

*10) The pattern followed by WVC and AOD (Figure 5.a) is different from the positive*

*correlation found by Huttuen et al. (2014).*

• Figure 5a illustrates the assumption made in the LUT approach, i.e. with increasing WVC, the retrieved AOD decreases for a given SSR. However, this assumption neglects a possible increase in e.g. aerosol hygroscopicity with increasing WVC which in turn would increase

AOD. The purpose of this figure is to point out how such a simpified assumption can cause a systematic bias in the LUT approach while machine learning techniques are not limited by such assumptions and can better constrain the effect of WVC on AOD.

*11) Figures 5 b and 5c show no clear differences between them.*

• The figure is updated and now the difference between Figures 5b and 5c is clearer due to the larger SZA range (48.5 to 51.5 degrees).

*12) In their analysis, the authors have used the single scattering albedo at 550 nm, but*

*in Figure 6 a they use the albedo for another wavelength, why?*

• The figure is updated. Now the wavelength is the same for both variables.

*13) Figures 6a and 6b should use the same scale for the same variable (water vapor*

*column) in order to enable comparison.*

• Figure 6a contains a subset from the whole data presented in Figure 6b and consequently, the

WVC axis was "zoomed" to improve readability.

*On the other hand, in Figure 6a the pattern shown for the albedo with WVC is different depending on*

*the interval considered for the WVC (slopes with contrary signs), thus there is no consistency between*

*Figures 6a and 6b because the pattern followed by WVC in Figure 6b is independent of the range*

*considered at WVC. It More discussion is necessary about the effect of water vapour,*

*considering other solar zenith angles for example.*

• In Figure 6a we had to select the measurements from a relatively small range of SZA and SSR, in order to demonstrate the physical reasoning behind the performance of LUT approach for a given input set of SSR, WVC, and SZA. In the Figure 6b, on the other hand, we wanted to include as much measurements as possible to show the general pattern of AOD vs. SSA relation and also the corresponding observed bias in LUT-estimated AOD as a function of WVC. For this reason, the range of x-axis was different. It is true that there is a WVC range when the SSA

to WVC slope differs from the overall pattern, and it happens below WVC of 2.5cm in the upper plot. However, it is arguably due to a limited amount of measurements in these bins, while the overall pattern is more important and causing the WVC dependent bias in LUT

approach that we wanted to demonstrate with the Figure 6.

*Concluding remarks: the paper can be accepted for publication after these comments*

*are taken into consideration and addressed.*

---

## Author Comment (AC2) · 2 May 2016

GENERAL COMMENTS

*I read the manuscript with interest, especially considering that it performs a comparison*

*of several multivariate techniques for modeling/estimating aerosol optical depth (AOD)*

*using surface solar radiation (SSR) measurements. As the authors point out, long time*

*series of such measurements are available and this can be exploited to reconstruct a*

*coincident record also of AOD. Extrapolation of AOD back in time is something that*

*will be very useful in studies of radiative forcing but also climate change trends. The*

*availability of long time series of AOD estimates will also help enrich models of other*

*atmospheric variables that would benefit from inclusion of this important parameter.*

*The study of AOD in the context of SSR is a very active field (a CrossRef metadata*

*search with +"aerosol optical depth" +"solar radiation" with the "journal article" flag*

*on returns a large number of 953,336 results), and it is good to see a study that is*

*targeted at AOD retrieval in particular. The authors idea of comparing machine learning*

*models is timely, well grounded and relevant to the scope of the journal of Atmospheric*

*Chemisty and Phyics (ACP). Several of the authors were instrumental in a recent ACP*

*paper to derive effective AOD from pyranometer measurements of SSR, by comparing*

*the capabilities of several modern approaches, the submitted manuscript builds on this*

*work and provides a useful feasibility study for the ballpark accuracy of AOD retrievals*

*from irradiances using advanced models.*

*Methodological issues:*

*1) On Page 4, lines 7-9, the authors describe how they have chosen to compare neural*

*network (NN), random forest (RF), Gaussian Process (GP) and Support Vector Ma-*

*chine (SVM) models of the AOD against look-up table (LUT) and nonlinear regression*

*models. Comparative studies of this type are becoming more popular in the literature,*

*but it should be born in mind that results are sensitive to model specification and, in*

*particular, the number of free parameters (e.g. Ljung, 1998). For example, in the con-*

*text of NN architectures alone, these include the number of neurons in hidden layers,*

*the number of such layers, training:validation data partition sizes, neuron activation*

*functions used). It is also rather challenging to find optimal values for model parame-*

*ters. For example, Meyer et al (2003) compared a SVM alone against 16 classification*

*methods and 9 regression methods in R. The same could be said for all of the methods*

*adopted in the submitted manuscript. With this in mind it would be good if the authors*

*could either:*

*a) increase the depth of the study by performing a thorough sensitivity analysis on the*

*free parameters used in each of the nonlinear modeling approaches (NN, RF, GP, SVM,*

*and NR) to help constrain the optimal values and number of free parameters needed*

*to achieve different model performance, or*

*b) emphasize more how the study performs a feasibility type of analysis of the specific*

*nonlinear models adopted for producing AOD retrievals of certain quality.*

• It is true that option a) would amount to an interesting study. Unfortunately, a sensitivity study which would constrain the optimal values and number of free parameters for different machine learning methods, in our opinion would amount to a whole new article.

In this study the aim is to validate methods that could be used for retrieving AOD, a proxy for aerosol load, for several decades.  As our study indicates, we get a good estimate for AOD with all of the machine learning methods used in this study and the study shows promise that these methods could be used for estimating past aerosol load. Thus our approach fall into category b) and we have emphasized in the revised manuscript that our study is more of a feasibility study.

*2) On Page 6, lines 8-11, the authors describe how the training dataset for the machine*

*learning methods contained years 2009-2014 and the validation (verification) dataset*

*contained the previous years 2005-2008. I would like to see the authors describe why*

*this partition was chosen (over others) as well as a short presentation of the basic*

*exploratory statistics of these datasets: i.e. the means and standard deviations and*

*min-max values of the model input and output parameters. This will help the authors*

*to make stronger claims about the generality of the models selected.*

• The main reason for choosing different time periods is that there may have been some change in the aerosol type between these two periods and this might cause problems for the methods to reproduce AOD's for one period when the learning data was from another period. Since the methods in this study are able to reproduce the AODs for a different time period than what they were trained for, it indicates that they have some capability in taking into account the changes in the aerosol type, i.e. change in the single scattering albedo.

We have also included some statistics on the data used, as the referee suggested. Table A1 shows the statistics between the training and validation datasets.

*SPECIFIC COMMENTS*

*I would say that the level of technical English in the submitted manuscript is reasonably*

*good, as is the level of scientific description. A couple of minor points:*

*3) On Page 3, lines 6-7, I disagree that AERONET has rather good spatial coverage.*

*Even on a global grid of 1 degree resolution (180 x 360 pixels), the occupancy of global*

*pixels, is extremely low dispite there being of the order of 10ˆ3 sites.*

• The referee is correct on this. We have rephrased this as follows: "Although, AERONET

contains globally already over 700 stations, with a fairly good spatial coverage compared to many other observation networks,"

*4) On Page 3, line 15, I would say that the (satellite and AERONET AOD) records*

*extend a between 1 and 2 decades into the past. On the daily timescale, this could be*

*arguably be considered to be a fairly long time-series record.*

- This is also correct. It now reads: " It is therefore apparent that neither sun-photometer nor satellite records of AOD are available for all decades where industrialization has had a significant effect on the aerosol load."

*5) I would make the font size bigger in Figure 1 and Figure 6.*

- This is fixed in the revised manuscript.

*6) In Figure 5, colour is associated with WVC and the title would be better placed ver-*

*tically on the colour bars as "WVC [cm] (LUT)" and "WVC [cm] (meas.)" or something*

*along these lines.*

- We agree with the referee and the colorbars' titles are now located at the top of the colorbars in

Fig. 5.We did not place them vertically next to the colorbars, as the referee suggested, because that would have made the figure harder to read.

*CONCLUDING REMARKS*

*Given the importance of accurate AOD estimation and the potenial for increasing the*

*capacity for monitoring long-term changes in climate forcing where AOD is a key pa-*

*rameter, the submitted manuscript is a useful addition to the literature and would benefit*

*I hope from these minor revisions.*

*REFERENCES*

*Ljung, L., 1998. System identification (pp. 163-173). Birkhäuser Boston.*

*Meyer, D., Leisch, F. and Hornik, K., 2003. The support vector machine under test.*

*Neurocomputing, 55(1), pp.169-186.*

*Interactive comment on Atmos. Chem. Phys. Discuss., doi:10.5194/acp-2016-58, 2016.*

---

## Author Response (AR2)

We appreciate the referee for these thorough fresh comments improving the paper more further. Below are the referee's comments followed by our replies:

*I agree with the changes cited in points 1, 3, 4, 5, 6, 8, 9, 10, 11 and 12 of my first revision.*

*My comments with respect to:*

*2) The method of Foyo-Moreno et al. (2014): the change made in the text by the authors is more*

*confusing than before.*

*Foyo-Moreno et al. (2014) finally proposed a method where a previous analysis which uses the neural*

*network concluded than the most adequate variable to estimate AOD is the ratio D/IN. This method*

*estimates AOD from solar radiation measurements, and in fact uses as the only input parameter a ratio*

*between diffuse radiation (D) and normal direct irradiance (IN) obtained from the difference between*

*the global irradiance (G) and diffuse irradiance (D), divided by the cosine of the solar zenith angle.*

*This should be mentioned explicitly and as separate reference, independent of the references of Olcese*

*et al. (2015) and Taylor et al. (2014), both of which use neural networks. This should be clarified.*

We changed the text from "For example, it has been applied to retrieve aerosol properties from remote sensing instruments (Olcese et al. 2015; Taylor et al., 2014; Foyo-Moreno et al, 2014)." to "For example, it has been applied to retrieve aerosol properties from remote sensing instruments (Olcese et al. 2015;

Taylor et al., 2014). Moreover, Foyo-Moreno et al, 2014 uses NN to indicate that a ratio between solar diffuse radiation and normal direct irradiance is the most adequate parameter to estimate AOD from solar radiation measurements."

*7) The errors associated to the fitting equation also should be included.*

We added the least square fits' errors into the caption text.

*Minor errors:*

*In Figure 5 the X axis should be changed to SSR consisting to the text.*

We changed Flux to SSR in the figure.